# Deciphering the Risk of Developing Second Primary Thyroid Cancer Following a Primary Malignancy—Who Is at the Greatest Risk?

**DOI:** 10.3390/cancers13061402

**Published:** 2021-03-19

**Authors:** Lily N. Trinh, Andrew R. Crawford, Mohammad H. Hussein, Mourad Zerfaoui, Eman A. Toraih, Gregory W. Randolph, Emad Kandil

**Affiliations:** 1School of Medicine, Tulane University, New Orleans, LA 70032, USA; ltrinh1@tulane.edu (L.N.T.); acrawfo2@tulane.edu (A.R.C.); 2Department of Surgery, Tulane University, New Orleans, LA 70032, USA; mhussein1@tulane.edu (M.H.H.); mzerfaoui@tulane.edu (M.Z.); 3Department of Histology and Cell Biology, Suez Canal University, 41523 Ismailia, Egypt; 4Department of Otolaryngology, Massachusetts Eye and Ear Infirmary, Boston, MA 02114, USA; grego-ry_randolph@meei.harvard.edu; 5Harvard Medical School Boston, Harvard Medical School, Boston, MA 02115, USA

**Keywords:** thyroid cancer, SPTC, breast cancer, systematic review

## Abstract

**Simple Summary:**

Associations between thyroid cancer and breast cancer have been elucidated, in that patients with breast cancer have a greater risk of developing subsequent thyroid cancer. However, not much is known about the relationship other primary cancers and subsequent thyroid cancer. In this review, we completed a thorough review of the existing literature to understand the relationship between primary cancers and second primary thyroid cancer (SPTC). Our findings suggest that surveillance protocols should be considered for patients at a higher risk of SPTC, including those with primary breast, renal cell, basal cell, and ovarian cancers who are female and/or Caucasian.

**Abstract:**

Background: It is critical to understand factors that may contribute to an increased risk of SPTC in order to develop surveillance protocols in high-risk individuals. This systematic review and meta-analysis will assess the association between primary malignancy and SPTC. Methods: A search of PubMed and Embase databases was completed in April 2020. Inclusion criteria included studies that reported the incidence or standardized incidence ratio of any primary malignancy and SPTC, published between 1980–2020. The PRISMA guidelines were followed and the Newcastle–Ottawa Scale was used to assess quality of studies. Results: 40 studies were included, which were comprised of 1,613,945 patients and 15 distinct types of primary cancers. In addition, 4196 (0.26%) patients developed SPTC following a mean duration of 8.07 ± 4.39 years. Greater risk of developing SPTC was found following primary breast (56.6%, 95%CI, 44.3–68.9, *p* < 0.001), renal cell (12.2%, 95%CI, 7.68–16.8, *p* < 0.001), basal cell (7.79%, 95%CI, 1.79–13.7, *p* = 0.011), and ovarian cancer (11.4%, 95%CI, 3.4–19.5, *p* = 0.005). SPTC patients were more likely to be females (RR = 1.58, 95%CI, 1.2–2.01, *p* < 0.001) and Caucasians (*p* < 0.001). Conclusions: Surveillance protocols should be considered for patients at a higher risk of SPTC, including those with primary breast, renal cell, basal cell and ovarian cancers who are female and/or Caucasian.

## 1. Introduction

There is a rising incidence of thyroid cancer in the United States [1]. This incidence has tripled in the last few decades and continues to rise at a rate of 4% per year [2]. Elevated rates of secondary primary thyroid cancer (SPTC) have been reported across a spectrum of primary cancer survivors [3,4]. By understanding the factors that may contribute to an increased SPTC risk, the underlying etiology of SPTC can be elucidated. Furthermore, the efficacy of surveillance protocols for high-risk individuals on the outcome of disease may be considered.

One emerging association with SPTC is breast cancer (BC) [5,6]. An et al. completed a retrospective case-control study demonstrating that the overall risk of SPTC or BC is increased in patients with prior BC or thyroid cancer, respectively [5]. Although the cause of this relationship remains unclear, potential factors for this association include chemotherapy and radiotherapy of the primary tumor, detection bias, genetic susceptibility, hormonal signaling, and environmental factors [6]. Connections between other primary malignancies such as stomach, liver, and colorectal cancers and SPTC have also been reported [7,8]. However, to date, there is no comprehensive review on the primary cancers associated with SPTC.

The primary aim of this meta-analysis is to evaluate the risk of developing SPTC following various primary malignancies. The secondary aim is to determine whether certain factors such as age, gender, race, and history of radiation and chemotherapy increase the risk of developing SPTC.

## 2. Materials and Methods

### 2.1. Literature Search Strategy

A comprehensive search was completed following the Preferred Reporting Items for Systematic Reviews and Meta-Analyses (PRISMA) guidelines. Titles and abstracts between 1980–2020, in the English language were searched across PubMed and Embase in April 2020. The medical subjects’ headings (MeSH) terms were used: (“primary malignancy” OR “cancer” OR “primary cancer”) AND (“secondary thyroid cancer” OR “thyroid cancer” OR “second primary thyroid cancer”). Articles including papillary, medullary, and follicular thyroid cancers were included. References of retrieved articles were also assessed for relevant studies.

### 2.2. Study Selection

Inclusion criteria included (1) any prospective or retrospective cohort study investigating the association of any primary malignancy and development of SPTC, (2) studies that reported the incidence or standardized incidence ratios (SIR) of cancers, and (3) studies published in a peer-reviewed journal. The major exclusion criteria were (1) abstract-only articles, conference articles, editorials, expert opinions, and (2) studies that did not stratify characteristics of various primary tumors.

### 2.3. Data Abstraction and Synthesis

Two (LNT and ARC) independent reviewers screened all citations for relevance and reviewed full-text articles. Any disagreement was resolved by discussion with another researcher (EAT). Data extraction was performed including relevant demographics, study design, publication year, and outcomes. The quality of each study was appraised by two independent reviewers (LNT and ARC) using the Newcastle–Ottawa Scale (NOS) [9].

### 2.4. Statistical Analysis

STATA Software version 16.0 (Stata Corporation, College Station, TX, USA) was used to perform a random-effects meta-analysis examining the SPTC risk based on incidence or SIR with the 95% confidence interval (CI) published in each study. Reported log risk ratio was converted to risk ratio. Hedges’ g was used to measure effect size. Potential heterogeneity across studies was examined using the Cochran’s Q-statistic and I^2^ statistic. Heterogeneity *p* < 0.05 or I^2^ > 50% indicates significant heterogeneity across studies. Meta-regression and subgroup analysis were employed to identify the source of heterogeneity.

## 3. Results

The search strategy identified 1525 citations yielding 1486 unique citations after exclusion of duplicates. Following exclusion of irrelevant articles, 111 full-text articles were reviewed. Workflow for study selection is demonstrated in Figure 1. Characteristics of studies are patients are depicted in Figure 2.

### 3.1. Characteristics of the Included Studies

A total of 40 articles (33 retrospective cohorts, 5 retrospective case-controls and 2 prospective cohorts) including 1,613,945 cancer patients with 15 distinct primary cancers were included in the analysis [5,7,10,11,12,13,14,15,16,17,18,19,20,21,22,23,24,25,26,27,28,29,30,31,32,33,34,35,36,37,38,39,40,41,42,43,44,45,46]. The study characteristics are shown in Table 1.

Breast cancer (BC) was the most frequent primary cancer (19 studies, 47.5%), followed by renal cell cancer (RCC) (3 studies, 7.5%) and basal cell carcinoma (BCC) (3 studies, 7.5%).

In total, 1,270,843 (78.74%) of the study population had breast cancer as the primary tumor site. Included studies were published in many countries (United States, India, China, Japan, Israel, etc.) in the last four decades (1984–2020). In addition, 1,480,816 (91.75%) women and 133,129 (8.25%) men were included in our study analysis. Quality assessment scores by the Newcastle–Ottawa Scale (NOS) are depicted in Table 2. Most of the included studies were comprised of low risk of bias, although some confounding was possible for studies that did not include any multivariable or otherwise adjusted analyses.

### 3.2. Characteristics and Risks of Any Second Primary Cancer Following a Primary Cancer

Across 40 studies, 96,675 (5.99%) survivors developed a second cancer different from their original primary cancer. In those who developed a second primary cancer, the most prevalent types of cancer were thyroid (37.1%, 95%CI, 20.8–53.2), lung (21.7%, 95%CI, 19.1–24.4), colon (21.2%, 95%CI, 15.7–26.6), and ovarian (14.8%, 95%CI, 9.6–20.0).

### 3.3. Characteristics and Risks of SPTC Following a Primary Cancer

SPTC was reported following 15 different types of primary cancer. 4196 (0.26%) survivors developed SPTC following a mean duration of 8.07 ± 4.39 years. Subgroup analysis by primary site revealed significant prevalence of thyroid cancer following BC (56.6%, 95%CI, 44.3–68.9, *p* < 0.001), RCC (12.2%, 95%CI, 7.68–16.8, *p* < 0.001), ovarian cancer (OC) (11.4%, 95%CI, 3.4–19.5, *p* = 0.005), and BCC of the skin (7.79%, 95%CI, 1.79–13.7, *p* = 0.011), Figure 3. Categorization of studies by date of publication revealed a high frequency of thyroid cancer in studies prior to 2015 (52.6%, 95%CI, 30.0–75.1, *p* < 0.001) compared to more recent articles (15.6%, 95%CI, 16.4–47.6, *p* = 0.33).

### 3.4. Characteristics of SPTC Patients

There was no significant difference between the age of patients who developed SPTC and those who were cancer free (Standard mean difference (SMD) = −0.02, 95%CI, −0.62–0.59, *p* = 0.96), Figure 4A. However, females were more likely to develop a SPTC (RR = 1.58, 95%CI, 1.2–2.01, *p* < 0.001), Figure 4B. Among the SPTC group, 81.3% were Caucasian and 9.2% were African Americans. Caucasian primary cancer patients were more than eight times more at risk to develop a second primary thyroid cancer (RR = 8.58, 95%CI, 7.02–10.48, *p* < 0.001), Figure 4C. In contrast, those who received chemotherapy (RR = 0.63, 95%CI, 0.26–1.47, *p* = 0.28) or radiotherapy (RR = 1.7, 95%CI, 0.03–81.8, *p* = 0.78) did not have a greater risk of developing SPTC.

## 4. Discussion

This current study is the largest multi-cohort analysis evaluating the risk of SPTC following any primary cancer. SPTC developed in 4196 (0.26%) of primary cancer patients following a mean duration of 8.07 ± 4.39 years. Patients were more likely to develop SPTC following primary BC, RCC, OC, and BCC of the skin. The highest risk of SPTC was found in patients with prior breast cancer. Caucasians and females were more likely to develop SPTC. Lastly, those who received chemotherapy or radiotherapy did not have a greater risk of developing thyroid cancer.

The most common primary cancer associated with SPTC was breast cancer. The relationship between breast and thyroid cancer has been previously studied. [6,32] However, the specific relevant risk factors and mechanisms remain unclear. One possible explanation for this relationship includes the hypothalamic-pituitary axis (HPA), which regulates both the mammary glands in breast and the thyroid gland. The increased actions of estrogen receptors in both glands have also been implicated [47]. As both thyroid and breast cancer are the most common cancers in women, a hormonal relationship between these cancers is not unsurprising. Of note, our study population comprised of 91.75% women with 78.74% of patients with primary breast cancer. Thus, the strong relationship between these cancers may be due to the bias of our study sample.

Our comprehensive study is the first review to report on various primary cancers associated with SPTC besides breast cancer. In particular, SPTC was associated with primary renal cell, ovarian, and basal cell skin cancers. The relationship between RCC and thyroid cancer has been questioned [48,49]. In a retrospective, population-based study, Fossen et al. found that female thyroid cancer patients had a 2-fold increase prevalence of subsequent RCC, whereas males had a 4.5-fold increase in subsequent RCC [48]. Key regulators such as circular RNA RAPGEF5 have been associated with both RCC and thyroid cancer [50]. Current literature on the relationship between these cancers sparse and more studies on their association should be further evaluated.

Existing data on the shared etiology of ovarian and thyroid cancer are limited. However, studies on this topic are emerging. In 2018, Weingarten et al. demonstrated that various transcription factors involved in the thyroid hormone axis may be associated with ovarian cancer metastasis [49]. In addition, thyroid hormones have been shown to bind to integrin αvβ3, which has oncogenic effects in cancers such as breast, ovarian, and melanoma [51]. Similar to breast cancer, there may be a hormonal pathway associated with these two cancers. The third primary cancer that showed a greater risk of SPTC was BCC of the skin [24]. Molecular studies have established that thyroid hormone action plays a critical role of multiple oncogenic pathways related to BCC tumorigenesis, specifically involving the miR21/GRHL3/D3 axis [24].

While not significant, other primary cancers associated with SPTC in this study includes leukemia and lymphoma, gastric cancer, and malignant melanoma. In a review by Schonfeld et al., patients who had chronic lymphocytic leukemia (CLL) and small lymphocytic leukemia (SLL) had an elevated risk of SPTC [52]. The authors suggest that thyroid dysfunction secondary to targeted therapy for lymphoid malignancies may be a potential explanation. Furthermore, studies have demonstrated shared genetic relationship between thyroid hormone and gastric and colorectal cancers. For example, Tamandani et al. demonstrated an association between promoter methylation and expression of thyroid hormone receptor beta (THRβ) gene in patients with gastric cancer in an Iranian population [53]. In addition, oncogenic mutations in BRAF, an oncoprotein, has been found in melanoma, thyroid cancer, and colorectal cancers [53].

Radiotherapy (RT) and chemotherapy during primary cancer treatment in childhood have been associated with the development of SPTC [4]. However, this association has not been supported in the adult population [20,54]. Sun et al. studied 55,318 women with breast cancer and found that the risk of thyroid cancer among women who received RT was not significantly higher than that of women who received no RT [54]. In addition, in a retrospective study involving 127,563 head and neck cancer survivors, those who received radiation were not at greater risk of developing SPTC [55]. Our study confirms that adults have a low risk of radiation-induced thyroid cancers. Studies involving the long-term effects of chemotherapy on SPTC are limited. In summary, many potential factors are related to the development of SPTC such as detection bias, genetic susceptibility, hormonal signaling, and environmental factors. Further studies on the relevant risk factors are needed to better understand these associations.

The finding of greater rates of thyroid cancer in certain populations must be evaluated in a broader context. One potential explanation for the higher risk of SPTC is detection bias. That is, since cancer patients are subject to increased surveillance following their first cancer diagnosis, they are more likely to be detected with a second primary cancer. Moreover, detection bias is associated with access to healthcare. Minority patients are less likely to have insurance and access to health care compared to Caucasian patients [56]. The more robust follow up of Caucasian patients may reflect the higher detection and thus association of SPTC in Caucasians compared with other races in our study. Finally, familial non-medullary thyroid cancer (FNMTC) accounts for approximately 5% of all thyroid cancers [57]. Patients with syndromes associated with FNMTC have a greater risk of developing cancers in addition to thyroid cancer. For example, familial adenomatous polyposis (FAP), a condition with multiple polyps in the colon, is associated with papillary thyroid cancer. Other examples include Cowden’s syndrome (associated with benign lesions of breast, colon, and endometrium, and brain) and Carney syndrome (associated with testicular, adrenal, or pituitary tumors) [57]. Due to our large study sample, our study may have included patients with such syndromes.

### Strengths and Limitations

This study has some strengths, including a comprehensive database search involving studies from all countries. In addition, we performed independent reviews of citations and articles in the selection of eligible studies, data abstraction, and critical appraisal of included studies.

There are a few limitations to this study. Due to the heterogeneity of study groups, there was a lack of data stratification by tumor type (i.e., papillary, follicular, medullary), staging, and recurrence of malignancy. Additionally, patient mortality was not included in our analysis. Other confounding factors such as smoking history, obesity, environment exposure, and family history of cancer were not reported in many of the included studies. Future studies should include an evaluation of these various factors when examining subsequent cancer risk.

## 5. Conclusions

An increased risk of SPTC was observed following a broad range of primary cancers. Surveillance protocols should be considered for patients at a greater risk for developing SPTC, including those with primary breast, renal cell, basal cell, and ovarian cancers who are female and/or Caucasian. Future studies should evaluate factors such as cancer staging and patient mortality.

## Figures and Tables

**Figure 1 cancers-13-01402-f001:**
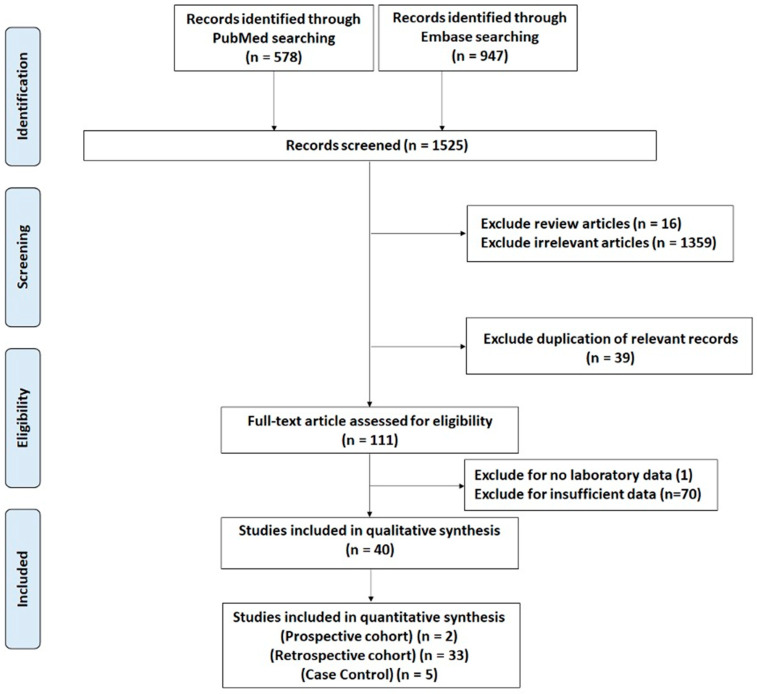
Systematic review process according to PRISMA guidelines.

**Figure 2 cancers-13-01402-f002:**
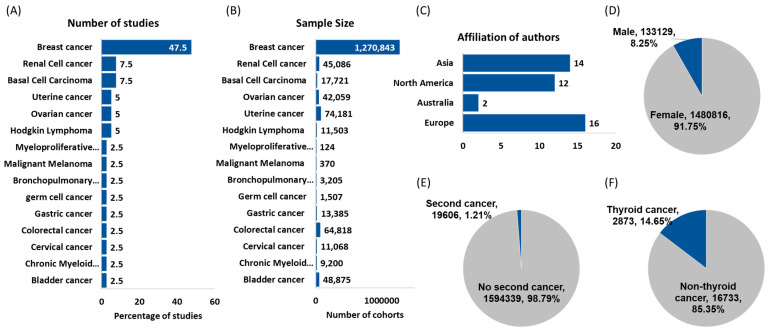
Features of studies and demographics of cancer patients. (**A**) distribution of studies according to the site of primary tumors; (**B**) sample size for each type of primary cancer; (**C**) geographical distribution of studies. Three studies were affiliated by authors from multiple countries; (**D**) distribution of cancer cohorts according to their sex; (**E**) frequency of a second primary cancer following another malignancy; (**F**) frequency of second primary thyroid cancer.

**Figure 3 cancers-13-01402-f003:**
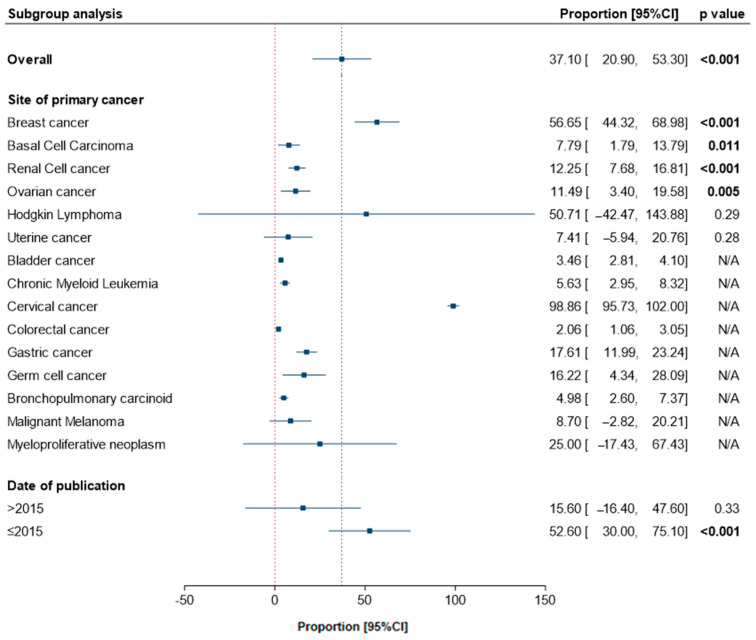
Pooled frequency of SPTC following a primary cancer. Overall and subgroup analysis are shown.

**Figure 4 cancers-13-01402-f004:**
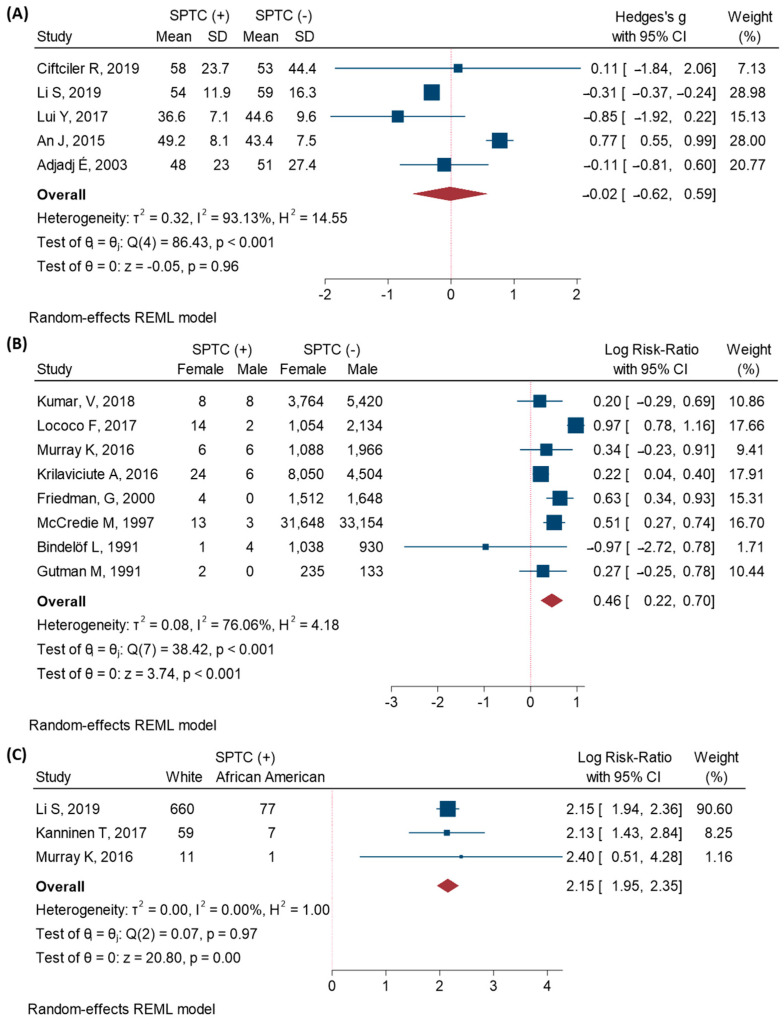
Characteristics of the SPTC patients. Random-effects model with restricted maximum-likelihood method was used. (**A**) Age, reported as standardized mean difference; (**B**) Gender, reported as log risk ratio; (**C**) Race, reported as log risk ratio.

**Table 1 cancers-13-01402-t001:** Characteristics of the study population.

Source	Country	Study Type	Study Period	Sample Size	Female	Male	Primary Cancer	Mean Age at Primary Diagnosis, Years	Follow-Up Duration, Years	SPTC ( + )	SPTC (−)
Schlosser S [10] (2020)	Israel	R	1991–2012	266	266	0	Breast	45.9	17	1	265
Ciftciler R [21] (2019)	Turkey	R	2001–2015	124	49	75	MPN	53.0	12	1	123
Li S [32] (2019)	USA	R	1992–2013	333,266	333,266	0	Breast	54.0	N/A	842	332,424
Bryk S [42] (2018)	Finland	R	1968–2013	986	986	0	Ovarian	40.0	N/A	6	980
Kwon W [43] (2018)	Korea	R	1993–2013	48,875	9524	39,351	Bladder	65.3	3.1	107	48,768
Corso G [44] (2018)	Italy	R	1994–2010	21,527	21,527	0	Breast	N/A	N/A	78	21,449
Joung J [45] (2018)	Korea	R	1993–2013	40,347	12,483	27,864	RCC	57.0	3.08	311	40,036
Kumar V [46] (2018)	USA	R	2002–2014	9200	3772	5428	CML	N/A	4.2	16	9184
Chen T [47] (2017)	Germany/Sweden	R	1997–2012	65,575	65,575	0	Uterine	68.5	3.9	42	65,533
Sud A [11] (2017)	Sweden	R	1965–2012	9522	4034	5488	HL	49.0	12.6	20	9502
Kanninen T [12] (2017)	USA	R	1992–2012	41,073	41,073	0	Ovarian	59.9	N/A	71	41,002
Lococo F [13] (2017)	Italy	R	1975–2011	3205	1069	2136	Lung	61.7	N/A	16	3189
Liao Z [14] (2017)	USA	R	1980–2011	1507	1507	0	Germ Cell	33.0	14.3	6	1501
Liu Y [15] (2017)	China	R	2008–2015	28	28	0	Breast	44.5	10.8	5	23
Murray K [16] (2016)	USA	P	1989–2010	3066	1972	1094	RCC	60.9	2.7	12	3054
Krilaviciute A [17] (2016)	Lithuania	R	1998–2007	12,584	8074	4510	BCC	N/A	14	30	12,554
Silverman B [18] (2016)	Israel	R	1990–2006	46,090	46,090	0	Breast	N/A	8.3	155	45,935
Marcheselli, R [19] (2015)	Italy	R	1996–2007	1830	1830	0	Breast	64.0	6.3	11	1819
An J [5] (2015)	Korea	R	1970–2009	6833	6833	0	Breast	43.4	4.4	81	6752
Michaelson E [20] (2014)	USA	R	1969–2008	1981	912	1069	HL	27.0	20.3	28	1953
Koivisto-Korander R [22] (2012)	Europe	R	1943–2000	8606	8606	0	Uterine	N/A	6.6	12	8594
Antonelli A [23] (2012)	Italy	R	1983–2009	1673	622	1051	RCC	61.6	5.9	15	1658
Tabuchi T [7] (2012)	Japan	R	1985–2004	13,385	4194	9191	Gastric	N/A	3.9	31	13,354
Yadav B [24] (2009)	India	R	1985–1995	1084	1084	0	Breast	N/A	12	1	1083
Lee K [25] (2008)	China	R	1979–2003	53,783	53,783	0	Breast	50.3	5.41	45	53,738
Kirova Y [26] (2008)	France	R	1981–1997	16,705	16,705	0	Breast	56.2	10.5	20	16,685
Mellemkjaer L [27] (2006)	Europe, Australia, Canada, Singapore	R	1943–2000	525,527	525,527	0	Breast	N/A	7.2	552	524,975
Sadetzki S [28] (2003)	Israel	R	1960–1998	49,207	49,207	0	Breast	N/A	7.1	72	49,135
Adjadj É [29] (2003)	France	R	1954–1983	200	200	0	Breast	48.0	9	8	192
Tanaka H [30] (2001)	Japan	R	1970–1995	2786	2786	0	Breast	50.9	8.6	7	2779
Huang J [31] (2001)	USA	R	1973–1993	194,798	194,798	0	Breast	N/A	N/A	140	194,658
Li C [33] (2000)	USA	R	1974–1994	2189	2189	0	Breast	59.8	4.2	20	2169
Friedman, G [34] (2000)	USA	R	1974–1997	3164	1516	1648	BCC	N/A	11.3	4	3160
McCredie M [35] (1997)	Australia	R	1972–1991	64,818	31,661	33,157	Colorectal	66.6	3.8	16	64,802
Volk N [36] (1997)	Slovenia	R	1961–1994	8791	8791	0	Breast	57.0	7.3	10	8781
Lindelöf B [37] (1991)	Sweden	R	1973–1983	1973	1039	934	BCC	68.0	6.5	5	1968
Gutman M [38] (1991)	Israel	R	1974–1986	370	237	133	MM	49.0	3.4	2	368
Boice J [39] (1988)	USA, Denmark, Sweden, UK	R	N/A	11,068	11,068	0	Cervical	52.0	N/A	43	11,025
Murakami R [40] (1987)	Japan	P	1965–1982	2786	2786	0	Breast	50.9	8.6	7	2779
Ron E [41] (1984)	USA	R	1935–1978	3147	3147	0	Breast	58.4	N/A	24	3123

R: Retrospective study, P: Prospective, SPTC: Second primary thyroid cancer, N/A: not applicable; MPN: Myeloproliferative neoplasms; CML: Chronic myelogenous leukemia; Lung: includes bronchopulmonary carcinoid HL: Hodgkin’s lymphoma; BCC: Basal cell carcinoma; MM: Malignant melanoma; RCC: Renal cell cancer.

**Table 2 cancers-13-01402-t002:** Quality index scoring based on Newcastle–Ottawa Scale (NOS).

Source	Selection (4 Max)	Comparability (1 Max)	Outcome (3 Max)	Overall Rating (8 Max)
Schlosser S [10] (2020)	2	1	2	5
Ciftciler R [21] (2019)	3	1	3	7
Li S [32] (2019)	3	1	2	6
Bryk S [42] (2018)	3	1	2	6
Kwon W [43] (2018)	3	1	3	7
Corso G [44] (2018)	3	1	2	7
Joung J [45] (2018)	3	1	3	7
Kumar V [46] (2018)	3	1	2	6
Chen T [47] (2017)	3	1	3	7
Sud A [11] (2017)	3	1	2	6
Kanninen T [12] (2017)	3	1	3	7
Lococo F [13] (2017)	3	1	3	7
Liao Z [14] (2017)	3	1	3	7
Liu Y [15] (2017)	3	1	2	6
Murray K [16] (2016)	3	1	3	7
Krilaviciute A [17] (2016)	3	1	3	7
Silverman B [18] (2016)	2	1	3	6
Marcheselli, R [19] (2015)	4	1	3	8
An J [5] (2015)	2	1	3	6
Michaelson E [20] (2014)	3	1	3	7
Koivisto-Korander R [22] (2012)	4	1	3	8
Antonelli A [23] (2012)	3	1	3	7
Tabuchi T [7] (2012)	3	1	2	6
Yadav B [24] (2009)	3	1	3	7
Lee K [25] (2008)	3	1	3	7
Kirova Y [26] (2008)	3	1	3	7
Mellemkjaer L [27] (2006)	3	1	3	7
Sadetzki S [28] (2003)	2	1	3	6
Adjadj É [29] (2003)	3	1	3	7
Tanaka H [30] (2001)	3	1	3	7
Huang J [31] (2001)	3	1	3	7
Li C [33] (2000)	3	1	2	6
Friedman, G [34] (2000)	3	1	3	7
McCredie M [35] (1997)	3	1	3	7
Volk N [36] (1997)	4	1	3	8
Lindelöf B [37] (1991)	3	1	3	7
Gutman M [38] (1991)	3	1	3	7
Boice J [39] (1988)	3	1	3	7
Murakami R [40] (1987)	4	1	2	7
Ron E [41] (1984)	3	1	2	6

## Data Availability

Data are available upon request.

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
