# Peer review of "Deciphering the Risk of Developing Second Primary Thyroid Cancer Following a Primary Malignancy—Who Is at the Greatest Risk?"

_cancers, 2021, doi:10.3390/cancers13061402_

Round 1
Reviewer 1 Report
Trinh et al. prepared systematic review and meta-analysis in order to assess the association between primary malignancy and secondary primary thyroid cancer (SPTC). They screened 1525 publications and identified 40 papers which included 1,613,945 cancer patients with 15 distinct primary cancers. These 40 papers were included in their quantitative synthesis. They found out that greater risk of developing SPTC was found following primary breast, renal cell, basal cell and ovarian cancer. Furthermore, SPTC patients were more likely to be females and Caucasians. The paper is clear and interesting and the discussion is relevant. The paper should be published in Cancers after a minor revision.
Minor remarks:
Line 55: a reference should be in square bracket
Line 86: a reference should be in square bracket
Table 1 (Line 1): Follow-up period is in years (not months)
Table 1: McCredie M (1997): a reference number is not 42 (it is 35)
Table 2: McCredie M (1997): a reference number is not 42 (it is 35)
Line 227: a reference [57] is missing
Author Response
Line 55: a reference should be in square bracket
Thank you for your comment. This reference has been corrected.
Line 86: a reference should be in square bracket
This reference has been corrected.
Table 1 (Line 1): Follow-up period is in years (not months)
The follow-up period in years has been corrected.
Table 1: McCredie M (1997): a reference number is not 42 (it is 35)
This reference number has been corrected.
Table 2: McCredie M (1997): a reference number is not 42 (it is 35)
This reference number has been corrected.
Line 227: a reference [57] is missing
This reference number has been added.
Reviewer 2 Report
In this review, the authors completed a thorough review of the existing literature to understand the relationship between primary cancers and second primary thyroid cancer (SPTC). The authors believe that surveillance protocols should be considered for patients at a higher risk of SPTC, including those with primary breast, renal cell, basal cell and ovarian cancers who are female and/or Caucasian.
This study is the largest multi-cohort analysis evaluating the risk of SPTC following any primary cancer. The paper is well documented, scientifically satisfactory and rigorous. The bibliography is good.
Author Response
In this review, the authors completed a thorough review of the existing literature to understand the relationship between primary cancers and second primary thyroid cancer (SPTC). The authors believe that surveillance protocols should be considered for patients at a higher risk of SPTC, including those with primary breast, renal cell, basal cell and ovarian cancers who are female and/or Caucasian.
This study is the largest multi-cohort analysis evaluating the risk of SPTC following any primary cancer. The paper is well documented, scientifically satisfactory and rigorous. The bibliography is good.
Thank you. We appreciate the response. There are no comments from this reviewer that required revisions in our manuscript.